# Mortality from Hypertrophic Cardiomyopathy in Brazil—Historical Series

**DOI:** 10.3390/ijerph21111498

**Published:** 2024-11-11

**Authors:** Emerson de Santana Santos, Pedro Henrique Gomes Castro, Laís Prado Smith Lima, João Victor Andrade Pimentel, Gabriel da Costa Kuhn, Antônio Carlos Sobral Sousa, Joselina Luzia Menezes Oliveira

**Affiliations:** 1Department of Medicine, Federal University of Sergipe (UFS), Aracaju 49100-000, Brazil; pedro.henriqueufs@bol.com.br (P.H.G.C.); laismith@academico.ufs.br (L.P.S.L.); jvapimentel@academico.ufs.br (J.V.A.P.); acssousa@terra.com.br (A.C.S.S.); joselina.menesesufs@bol.com.br (J.L.M.O.); 2Postgraduate Program in Health Sciences, Federal University of Sergipe (UFS), Aracaju 49060-676, Brazil; 3University Hospital, Federal University of Sergipe, Aracaju 49100-000, Brazil; 4Department of Medicine, Tiradentes University (UNIT), Aracaju 49032-490, Brazil; gabriel.costa02@souunit.com.br; 5Division of Cardiology, University Hospital, Federal University of Sergipe (UFS) , Aracaju 49100-000, Brazil; 6Clinic and Hospital São Lucas/Rede D’Or São Luiz, Aracaju 49060-676, Brazil; 7Hospital Primavera, Aracaju 49060-010, Brazil

**Keywords:** hypertrophic cardiomyopathy, mortality, trends

## Abstract

Hypertrophic cardiomyopathy (HCM) is a relatively prevalent disease, primarily of a genetic etiology, affecting both sexes and characterized by left ventricular hypertrophy. However, limitations within healthcare systems, socioracial factors, and the issue of underdiagnosis hinder accurate mortality assessments in our region. This study, therefore, aimed to assess the mortality trends associated with HCM in Brazil from 2010 to 2020, with a focus on socioracial factors and healthcare disparities. This ecological, time-series study employed a quantitative approach based on secondary data from the Mortality System (SIM) developed by the Brazilian Ministry of Health. Mortality incidence and trend analyses were conducted using the average annual percent change (AAPC) and the annual percent change (APC). The results indicated a predominance of HCM-related deaths among white males aged 40 years and older. Additionally, an increasing trend in HCM-related mortality was observed among white and brown males and females aged 40 years and above from 2010 to 2018. Throughout the entire period covered in the study, the incidence of deaths due to HCM increased by 18.3% and 69.8% in the northeastern and southeastern regions. The findings suggest that health system managers should consider addressing the factors influencing HCM mortality and encourage the development and implementation of clinical protocols across healthcare institutions nationwide. Such protocols are recommended to facilitate early diagnosis and establish effective treatment strategies, ultimately aiming to improve the survival rates and quality of life for individuals affected by HCM.

## 1. Introduction

Hypertrophic cardiomyopathy (HCM) is an autosomal dominant hereditary heart disease with diverse phenotypic expression, widespread worldwide and affecting both sexes and individuals with different races and ethnicities [1], being the most common cardiovascular disease with a genetic etiology. Its prevalence ratio is between 1:200 and 1:500 and its mortality is associated with sudden cardiac death and heart failure [1].

The disease causes left ventricular hypertrophy and disorder of the myocyte architecture with the development of cardiac muscle fibrosis, which results in impaired diastolic function, obstruction of the left ventricular outflow tract and consequent heart failure and arrhythmias, such as atrial fibrillation [2].

Mutations mostly affect genes that encode proteins involved in sarcomeric contraction, of which the most commonly involved are beta myosin heavy chain (MYH7), myosin binding protein C (MYBPC3) and troponin T (TNNT 2) [3]. The presence of phenocopies is also typical of the disease and is characterized by the phenotypic expression of HCM, but without its genetic mutations. Cardiac amyloidosis, Fabry disease and the cardiomyopathies LAMP2 and PRKAG2 are examples of similar phenotypes [4].

HCM is considered the most common cause of sudden cardiac death in competitive athletes, resulting from fatal ventricular arrhythmias [5], and in young individuals under the age of 30. The risk generally extends into middle age and reduces in patients over the age of 60 [6]. However, a large proportion of affected individuals do not develop symptoms, which favors underdiagnosis [2].

Despite these challenges, significant progress in the management of HCM has been made in recent decades. Advances in the management of this disease have allowed changes in its natural history, promoting a reduction in the mortality rate and an improvement in the quality of life of patients [7]. The implantation of modern cardioverter defibrillators is one of the explanations for the increased survival of patients with this heart disease [8].

It is estimated that there has been a reduction in mortality from 6%/year to 0.5%/year, with this remaining rate being attributed to individuals with progressive refractory heart failure [1]. Meanwhile, when evaluating children, an annual mortality rate of around 1% is identified [9]. Sophisticated cardiac imaging methods, timely diagnosis from index cases, rhythm monitoring and the use of appropriate pharmacotherapy, together with cardioverter defibrillators and advances in risk stratification, have contributed to the progressive reduction in mortality from the disease [2].

Given the relevance of this disease and considering its high incidence, even when underdiagnosed, the objective of the present study was to evaluate the mortality due to HCM in Brazil from 2010 to 2020. This analysis includes the association of sociodemographic data related to mortality rates, as understanding these factors is crucial in identifying disparities in health outcomes across different populations [8].

Sociodemographic factors, such as socioeconomic status, regional social vulnerability and healthcare accessibility, significantly influence both the diagnosis and outcomes of HCM. Individuals from lower socioeconomic backgrounds may experience reduced access to healthcare services, leading to delays in diagnosis and inadequate management of the disease. Furthermore, regions with higher social vulnerability often exhibit lifestyle-related risk factors and lower treatment adherence, which can exacerbate the severity of HCM and increase the mortality rates. By examining these factors, this study aims to provide valuable insights that can support future research and inform the formulation of public policies designed to promote early diagnosis strategies, ultimately improving health outcomes in diverse communities [10].

## 2. Materials and Methods

This was an ecological, time-series study with a quantitative approach based on data from the Mortality System (Sistema de Informações sobre Mortalidade; SIM) of the Unified Health System (Sistema Único de Saúde; SUS), obtained through access to the Department of Information Technology of the Brazilian Ministry of Health (DATASUS).

Data entry to the SIM is accomplished through the mandatory completion and submission of the Epidemiological Death Certificate, an official document established by the Brazilian Ministry of Health and in use since 1976. This certificate is exclusively filled out by the attending physician and includes information on patient identification, sociodemographic data, clinical history and details regarding the conditions and cause of death, including the ICD code. This standardized process helps to minimize temporal and spatial discrepancies in data entry.

As this dataset is publicly accessible, contains no personal identifiers and does not compromise participant confidentiality, this study was exempt from review under National Health Consul Resolutions 466/2012 and 510/2016 for registration and analysis by ethics committees in research involving human beings.

The sample for this study comprised records of deaths due to hypertrophic cardiomyopathy (HCM), identified by the ICD I42.2, on the SIM platform between 2010 and 2020, categorized by sex, age group (0–19, 20–39, 40–59, 60–79, and 80+ years), region and race. The classification of these variables adhered to the criteria established by the Brazilian Institute of Geography and Statistics (IBGE), which organizes the Brazilian population into five distinct geographical regions: north, northeast, central–west, southeast and south. This classification takes into account various factors, including geography, culture and economy. Additionally, the institute categorizes its population into five racial and ethnic groups: white, black, brown (Pardo), yellow (Asian), and Indigenous.

The variable “ignored” was also added to indicate the absence of this information in the source document. All individuals of both sexes and of all age groups, with a confirmed diagnosis of the disease documented in the SIM between 2010 and 2020, were included. Missing or incomplete data were excluded from the analysis.

Data were tabulated using the TABWIN 4.15 (DATASUS) software, allowing for the systematic extraction of mortality records based on predefined inclusion criteria. These tabulated data were subsequently exported to Microsoft Excel 2019. Within Excel, the data were organized by year, demographic factors and other study variables. The incidences of specific causes of death were modeled using a generalized linear model (GLM) with a Poisson distribution estimated by the quasi-likelihood method. The absence of different angular coefficients, i.e., that there were no change points in the slope of the curve of the incidences of specific causes of death, was tested using the (pseudo-)score test. The objective of the pseudo-score test is to verify whether there is sufficient statistical evidence to reject the null hypothesis that the time series or curve is linear and does not present abrupt changes in its trend or behavior and to accept the alternative hypothesis that there are significant inflection points in the time series or curve.

The annual percent change (APC) and average annual percent change (AAPC) were calculated to identify trends in mortality over time, with the APC estimated for each segment if inflection points were detected. Statistical significance was set at *p* ≤ 0.05 and the software used was R Core Team 2022 (Version 4.2.2). The figures, starting from the red segment, expressed the associated coefficients. When there was an inflection point in the trend, the APC was generated for each period before and after the inflection.

## 3. Results

From 2010 to 2020, a total of 141,213 deaths were recorded due to cardiomyopathies, of which 25,327 (17.9%) were specifically attributed to HCM. Figure 1 shows the incidence trend, with an increase between 2010 and 2018 (APC of 3.9% per year) and a decline between 2019 and 2020 (APC of −95.1%), resulting in an overall AAPC of −7.2%.

Among these deaths, 15,723 (62%) occurred in men. The overall mortality trend indicated an increase in deaths among both men and women from 2010 to 2018, followed by a notable decrease from 2019 to 2020. Specifically, the incidence among men increased at a rate of 5.8% per year, while a significant decline of −10.7% per year was observed during the latter period. Conversely, among women, the incidence increased by 6.5% per year, followed by a decrease of −10.7% per year. Ultimately, the overall trend for both genders reflected a period of stability, with average annual percentage changes (AAPCs) of −1.1% for females and −1.3% for males. These trends are further illustrated in Figure 2 and Figure 3, which display segmented patterns of HCM-related mortality for men and women, respectively. Both figures highlight distinct periods of growth and decline, with an overall AAPC of −6.8% for men (Figure 1) and −7.8% for women (Figure 2) throughout the study period.

In terms of age distribution, individuals aged 40 years or older accounted for 88% of the deaths attributed to hypertrophic cardiomyopathy (HCM) during the study period, with the age group of 60 to 79 years being the most affected, representing 38.8% (9849 deaths) of the total. Mortality trends among individuals aged 40 and above showed annual growth rates of 4.8% for those aged 40 to 59, 6.6% for those aged 60 to 79, and 6.9% for those aged 80 or older from 2010 to 2018. In contrast, no significant trends were observed for individuals aged 0 to 19, indicating stability in mortality within this younger age group. Throughout the 2010–2020 period, a similar trend was evident among individuals aged over 40, except for the 20 to 39 age group, which experienced an average annual growth rate of 4% (AAPC). This is further illustrated in Figure 4 and Figure 5. Figure 4 shows the incidence of HCM-related deaths per thousand inhabitants for individuals aged 0 to 19 years, with a significant increase observed between 2010 and 2017 (APC of 11.3% per year). Figure 5 presents the data for individuals aged 20 to 39 years, where a growth of 16% per year was observed from 2010 to 2016, followed by a decline of −9.2% per year from 2017 to 2020, resulting in an overall AAPC of 5.7%.

The distribution of deaths by race revealed a predominance of white individuals, who accounted for 11,873 deaths (46.8%), followed by brown individuals with 10,164 deaths (40.1%) and black individuals with 2799 deaths (11%). Additionally, individuals of Asian, Indigenous or unknown ethnicity represented a small portion of the total, comprising 491 people (1.9%). The trends among white and brown individuals displayed a similar pattern, with an increase in mortality until 2018, followed by a decline from 2019 to 2020. Specifically, the number of deaths among white individuals increased at an annual rate of 1.1% until 2018 and then decreased significantly by −14.3% between 2019 and 2020, resulting in an overall plateau (AAPC of 1.2%). Brown individuals also experienced an increasing trend of 4.6% per year until 2018, followed by a significant decline of −19% per year. Meanwhile, the number of deaths among black individuals rose at an annual rate of 11% until 2015 but subsequently decreased by −1.5% from 2016 to 2020, resulting in an AAPC of 1.9%.

Geographically, the southeastern region recorded the highest number of deaths due to HCM, totaling approximately 17,684 deaths (69.8%). This was followed by the northeast, which reported 4,640 deaths (18.3%), the south with 1,832 deaths (7.2%), the midwest with 895 deaths (3.5%) and the north with 276 deaths (1%). Notably, the northern region exhibited a continuous decrease in mortality over the entire period, with an annual decline of −1.9%. In contrast, the southern and southeastern regions experienced increases in their mortality rates of 2.6% and 4.1% per year, respectively. Meanwhile, the central–western and northeastern regions showed little to no change, with annual percentage variations of −1.9% and 0.8%, respectively.

## 4. Discussion

The present study analyzed HCM-related deaths, addressing both sexes, ages and race across the five regions of Brazil. However, it is important to clarify that, as an ecological and temporal study, it may present some limitations, such as underdiagnosis.

A significant limitation of this study stems from the inability of the ICD code I42.2, designated for hypertrophic cardiomyopathy, to distinguish HCM-related mortality from deaths due to other heart disorders with similar phenotypes, known as phenocopies. Diseases such as Fabry disease, cardiac amyloidosis and hypertensive cardiomyopathy, which can also present with cardiac enlargement, may be misclassified as HCM. This issue is exacerbated by the reliance on imaging for diagnosis without an accompanying genetic investigation, a common limitation in developing countries like Brazil, where limited access to healthcare services hinders accurate diagnosis. Furthermore, the scarcity of international studies with comparable scope to this work restricted the opportunities for comparative and observational analyses.

According to Maron et al. (2018) [11], approximately 340,000 deaths per year from HCM occur globally, with a mortality rate per 100,000 individuals of 5.6 for men and 3.3 for women. In a cohort study conducted in a hospital in São Paulo, the mortality rate was 1%, which is low considering that the study was conducted in a reference center [12]. In the present study, the total number of deaths from the disease in males was higher compared to females.

However, there was a greater increase in the incidence of deaths in women during the period from 2010 to 2018, with a rate of 6.5%/year, compared to 5.8%/year in men. According to van Driel et al. (2019) [13], women with hypertrophic cardiomyopathy (HCM) are often diagnosed at an older age and present with more advanced disease, characterized by greater left ventricular outflow tract obstruction and more severe symptoms compared to men. This delay in diagnosis and treatment likely contributes to poorer prognoses among female patients, emphasizing the need for earlier recognition and intervention. As highlighted by Geske et al. (2018) [14], these gender-based differences in the clinical course underscore the importance of diagnostic strategies that can detect HCM in women at earlier stages, thereby improving the health outcomes in this population.

The predominance of deaths in the male population was also observed in the study by Olivotto et al. (2005) [15], who found this same trend when evaluating research related to the topic. However, it is still unknown whether there is lower penetrance of the disease in women or failure in the recognition and clinical referral of affected patients.

A growing trend in the incidence of deaths in women was observed by Geske et al. (2017) [16] based on a study involving 3673 adults, with 45% of the female patients followed for a period of 16 years, in which women had a significantly higher incidence of mortality. In contrast, Peeling et al. (2017) [17] point out that the female population in Brazil, in the age groups below 60 years old, has lower mortality rates than the male population, both overall and due to specific causes. An American study carried out with 2123 patients (38% women) between 2001 and 2006 showed no difference in the incidence of mortality between both sexes [18].

Javidgonbadi et al. (2021) [19] further explored the increasing trend in female mortality associated with hypertrophic cardiomyopathy, identifying several contributing factors to this disparity. Their study reveals that excess female mortality primarily occurs in cases of heart failure and acute myocardial infarctions among age-matched groups. Notably, women present with more severe septal hypertrophy relative to the body surface area (BSA) at diagnosis, yet often receive less aggressive treatment, including lower rates of beta-blocker and disopyramide therapy. The authors suggest that delayed diagnosis could also contribute to the increased mortality in females. Additionally, they highlight that sex bias in diagnostic criteria and disparities in healthcare access may significantly impact outcomes.

Another European study, which recruited 4893 patients (36% women) who were followed for 6.2 years, showed a higher incidence of death due to HCM among women persisting until the last few decades of life, while the incidence of death in men over 65 years of age was similar to that of the general population [20]. This finding may be related to deaths due to heart failure, since this dysfunction is more present in females when associated with hypertrophic disease [21]. Furthermore, Bhalla et al. (2022) [22] observed that women are, on average, 9 years older than men at the time of diagnosis and have more symptoms of heart failure due to outflow tract obstruction or diastolic dysfunction, which may explain the increase in the incidence of death in this population group.

When evaluating the proportions of deaths by race, there was a predominance of white individuals (46.8%), who accounted for almost half of the deaths between 2010 and 2020, followed by brown individuals, with a significant percentage (40.1%), and black individuals (11%). According to Eberly et al. (2020) [23], the low proportion of black patients in the cohorts suggests inadequate referral to specialized care. Wells et al. (2018) [24] explain the low proportion of black patients based on under-referral and a lack of accessibility to specialized services for this population group, associated with racial inequalities in referral patterns. The study demonstrates a decline in the incidence of HCM-related mortality among the black population starting in 2015. According to Silva et al. (2017) [25], the coverage of healthcare services and the utilization of medical procedures have increased among the black population, reflecting heightened awareness of the need for equity. Family health initiatives and primary care programs contributed to the improvement in health indicators.

Arabadjian et al. (2021) [26] observed, after analyzing multiple cohorts involving deaths from HCM, that the participation of black people rarely exceeded 10% representation. However, when analyzing the incidence, black individuals showed greater growth than any other race, in the period of 2010 to 2015, while white and brown individuals had more modest growth between 2010 and 2018.

A retrospective study that collected 39,200 deaths from HCM between 1999 and 2019 in the United States showed that black individuals in the period from 2012 to 2019 experienced an increase in the incidence of death from HCM, while white individuals and other races showed a decline in the same period. Minhas et al. (2021) [27] attribute this increase to disparities in healthcare, lower rates of genetic testing, the use of implantable cardioverter defibrillators and septal reduction therapy in this group of individuals. According to Arabadjian et al. (2021) [26], black people are more prone to the development of hypertrophy and left ventricular obstruction, in addition to being more susceptible to the development of NYHA III and IV heart failure.

Minhas et al. (2021) [27] identified higher mortality among individuals over 75 years of age in the period from 1999 to 2019, even with a reduction in the mortality rate throughout the period. Alashi et al. (2021) [28] attribute this trend of higher mortality among older individuals to cardiovascular risk factors, including arterial hypertension, which are more prevalent in this population than the risk factors for sudden cardiac death.

Messner et al. (2003) [29] observed that the mortality rate for sudden cardiac death (SCD) increased with increasing age after evaluating the results of a 15-year cohort in Sweden that showed an annual incidence of SCD of 8 per 100,000 men in the age group of 35–44 years and 171 per 100,000 men in the age group of 55-64 years. Hua et al. (2009) [30] also found this result when analyzing the incidence of SCD in China, where there was a significant increase in the number of deaths with advancing age, with the majority of deaths occurring in individuals aged 65 years or older.

When analyzing the incidence, a growing trend was observed between 2010 and 2018 in the age group of 40 years old or older. In the group aged 20 to 39 years old, growth was observed throughout the period, while, among young individuals aged 0 to 19 years old, no trends were identified and there was stagnation. Between 2019 and 2020, there was a sharp drop in incidence, especially in the group of individuals over 40 years old.

Despite fluctuations in specific periods, the overall trend remained relatively stable over the decade. The stagnation in the number of HCM-related deaths among young individuals in the last decade can be attributed to significant advancements in diagnostic techniques, including echocardiography and cardiac magnetic resonance imaging. According to Canciello et al. (2024) [31], these innovations have facilitated the early identification of the condition in asymptomatic individuals, resulting in more effective management. Additionally, increased awareness of HCM, particularly among athletes, has led to more frequent screenings, enabling the detection and treatment of high-risk cases before the occurrence of adverse events, as noted by Corrado et al. (1998) [32]. Surgical interventions, such as septal myectomy, have also demonstrated effectiveness in reducing HCM-related mortality [33]. Therefore, the combination of early diagnoses, heightened awareness, screening initiatives and effective treatments likely explains the stagnation in the mortality rates due to hypertrophic cardiomyopathy in the young population.

The downward trend in the incidence of death from hypertrophic cardiomyopathy between 2019 and 2020 could be explained by the convergence with the period of the COVID-19 pandemic. The underreporting of deaths from cardiovascular diseases during the health crisis has been noted. This is corroborated by a study developed by Normando et al. (2021) [34], who analyzed the impact of the pandemic on the number of hospitalizations and hospital deaths due to cardiovascular etiologies in a comparative study between the year 2020 and the period from 2016 to 2019.

Additionally, a study by Ssentongo et al. (2021) [35] reported that medical resources were predominantly allocated to COVID-19 cases, resulting in delays and cancellations of non-COVID-19 elective procedures and routine care, which, in turn, increased the morbidity and mortality among patients requiring ongoing management for chronic conditions such as hypertrophic cardiomyopathy.

Moreover, many causes, including the fear of contracting COVID-19, may have discouraged many individuals from seeking medical care [36], which can have led to delayed HCM diagnoses and an increase in sudden cardiac events, some of which may have been misclassified as COVID-19-related or unexplained deaths [37].

The author observed a 15% reduction in hospitalizations and a 9% reduction in hospital deaths, in addition to finding a 9% increase in the in-hospital mortality rate for this group of diseases. Alves et al. (2020) [38], in a study developed in the state of Minas Gerais, observed that, in relation to the same period in 2019, there was an increase of 18.44% in deaths due to cardiovascular etiologies in the home environment in 2020, excluding stroke and acute myocardial infarction. According to Normando et al. (2021) [34], this trend of increasing home deaths can be explained by the restructuring of the dynamics of hospital care and by the reduction in the population’s demand for care due to fear of infection due to SARS-CoV-2. This may be one of the explanations for the reduction in the incidence of death from HCM.

Alashi et al. (2021) [28], in their study, did not find significant variation in the incidence of death in the groups between 60 and 74 years old, in which stagnation was observed.

A Portuguese cohort study involving 4591 patients with the disease, who were followed for 5.4 years, showed that there were two main predictors of death in patients, which were sarcomeric mutation and the age at diagnosis. Regarding age, the mortality in young people (20 to 29 years) was four times higher than in the general population adjusted for age, and, in patients between 50 and 69 years old, it was three times higher, with the main causes of death being heart failure and non-cardiac death, while sudden cardiac death occurred in only 16% of these deaths. Regardless of age, most complications related to HCM occurred late, between 50 and 70 years of age, due to atrial fibrillation and heart failure [39].

Regarding the total distribution of deaths, the southeastern and northeastern regions stood out, with 88.1% of the deaths in the period of 2010 to 2020. Leal et al. (2018) [40] explain this finding based on the greater population concentration and higher absolute notifications of deaths. This phenomenon can also be explained in light of the Social Determinants of Health (SDH), since, historically, the northeastern region has higher rates of poverty, low educational inclusion and difficulties in accessing health services [41]. Such factors contribute to late diagnoses with worse outcomes.

The greater concentration of HCM-related deaths in the southeastern and northeastern regions of Brazil can be better understood by examining the healthcare infrastructure and socioeconomic factors unique to these areas. According to Oliveira et al. (2018) [42], the southeastern region, which includes states such as São Paulo and Rio de Janeiro, possesses advanced healthcare facilities, greater access to specialized cardiology services and a greater concentration of medical professionals. This can lead to more accurate diagnoses and proper notification of deaths. Conversely, the northeastern region faces significant challenges, including limited access to high-quality healthcare services, fewer specialized cardiologists and economic disparities that hinder patients from receiving accurate diagnoses and necessary medical care.

Additionally, it was noted that the north had the lowest total number of deaths due to HCM during the entire period. In addition to the fact that this region is less populated, the issue of underreporting may have contributed to this finding. Braggion-Santos et al. (2015) [43], in their study carried out in Ribeirão Preto, found a shortage of records and observed that only 3.6% of the forms completed by doctors and 2.2% of the autopsy reports had SCD as the cause of death.

Timerman et al. (2006) [44] found a similar result when evaluating data from the Program for the Improvement of Municipal Mortality Information (Programa de Aprimoramento das Informações de Mortalidade do Município; PRO-AIM) in the Municipality of São Paulo, in which 62,895 deaths occurred in 1998 (21,044 due to cardiovascular disease), of which 38.7% were secondary to coronary disease; however, only one case of sudden death was reported in the death records for that year.

## 5. Conclusions

This study reveals an upward trend in hypertrophic cardiomyopathy (HCM) mortality from 2010 to 2018, predominantly affecting white and brown individuals aged 40 years or older, with a sharper increase observed in women during this period. The highest mortality rates were identified among white men over 40, especially in Brazil’s northeastern and southeastern regions. However, the notable decrease in HCM mortality from 2019 to 2020 suggests that the COVID-19 pandemic may have influenced cardiovascular death reporting, potentially leading to the underreporting of HCM-related deaths.

These findings underscore the importance of targeted public health interventions to address specific populations with higher HCM mortality, such as women, certain racial groups and individuals in high-prevalence regions. Efforts should prioritize raising awareness, enhancing early detection and improving access to specialized treatments within these communities. Allocating resources to expand healthcare accessibility and provide education on HCM would help to reduce disparities, decrease the mortality rates and improve patient outcomes.

Moreover, the pandemic highlights the critical need for data accuracy during public health crises. To improve the data reliability, it is essential to modernize reporting systems, incorporate real-time surveillance technologies and strengthen interagency collaboration for efficient data sharing. Comprehensive training for healthcare professionals on data management and the importance of precise documentation could further enhance the integrity of health records. These measures will improve the quality of epidemiological research and inform effective public health responses.

## Figures and Tables

**Figure 1 ijerph-21-01498-f001:**
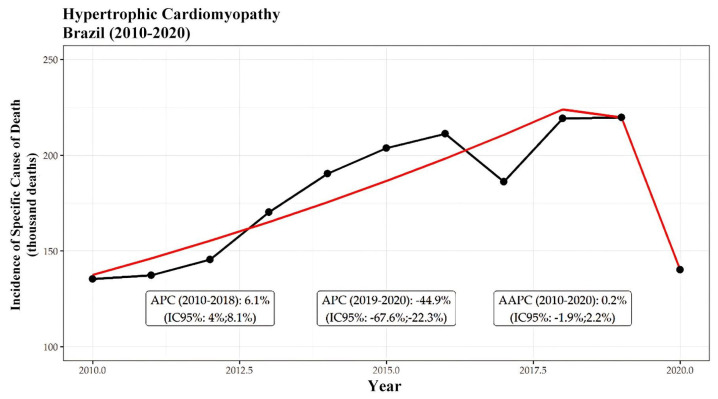
The black line represents the incidence of hypertrophic cardiomyopathy, while the red line reflects the segmented model, highlighting the periods of growth and decline. Between 2010 and 2018, an increase was observed with an APC of 3.9% per year, followed by a decline between 2019 and 2020, with an APC of −95.1%. The overall incidence showed an AAPC of −7.2%.

**Figure 2 ijerph-21-01498-f002:**
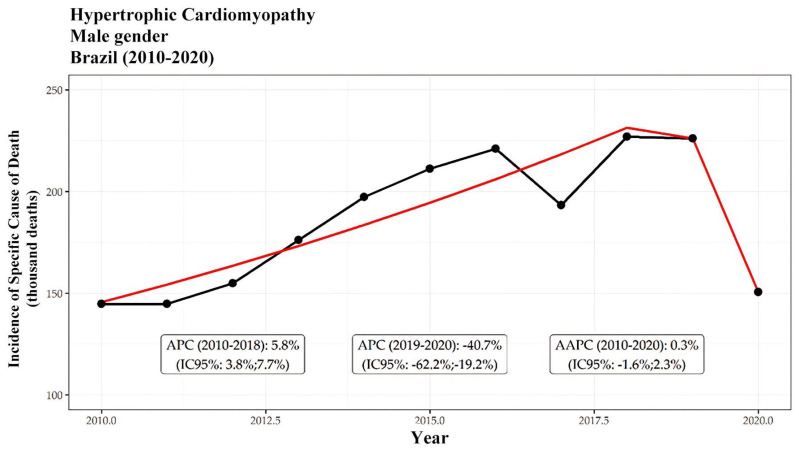
The incidences of HCM-related death for thousand male are presented. The black line represents the incidence, while the red line reflects the segmented model, highlighting periods of growth and decline. Two statistically significant trends are observed: growth between 2010 and 2018 with an APC of 3.4% per year, and a decline between 2019 and 2020 with an APC of −93.9%. Overall, the incidence decreased with an AAPC of −6.8%.

**Figure 3 ijerph-21-01498-f003:**
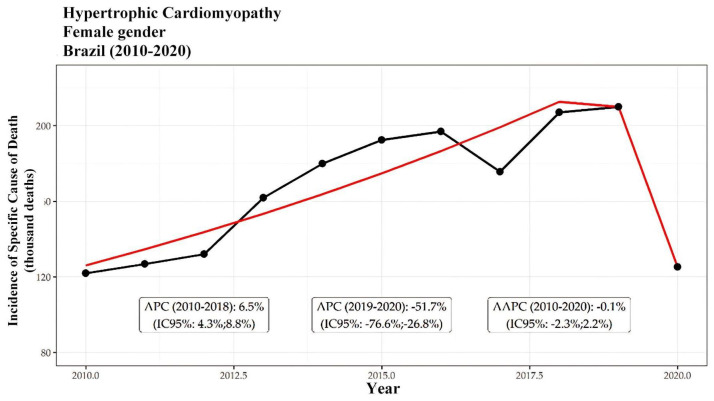
The incidences of HCM-related death for every thousand female are presented. The black line represents the incidence, while the red line reflects the segmented model, highlighting periods of growth and decline. Two statistically significant trends are observed: growth between 2010 and 2018 with an APC of 3.4% per year, and a decline between 2019 and 2020 with an APC of −97.8%. Overall, the incidence decreased with an AAPC of −7.8%.

**Figure 4 ijerph-21-01498-f004:**
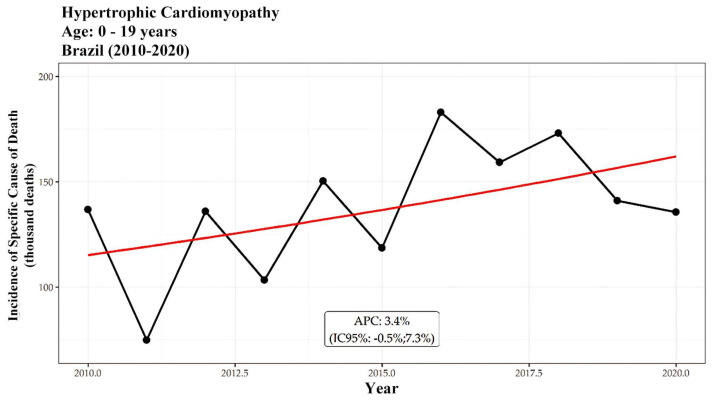
The incidences of specific cause of death per thousand inhabitants related to hypertrophic cardiomyopathy are presented for patients aged 0 to 19 years. The black line represents the incidence, while the red line reflects the segmented model, which generates the APC and AAPC. APC, or annual percent change, indicates the average variation each year. It can be observed that there is one statistically significant trend: growth between 2010 and 2017 with an APC of 11.3% per year.

**Figure 5 ijerph-21-01498-f005:**
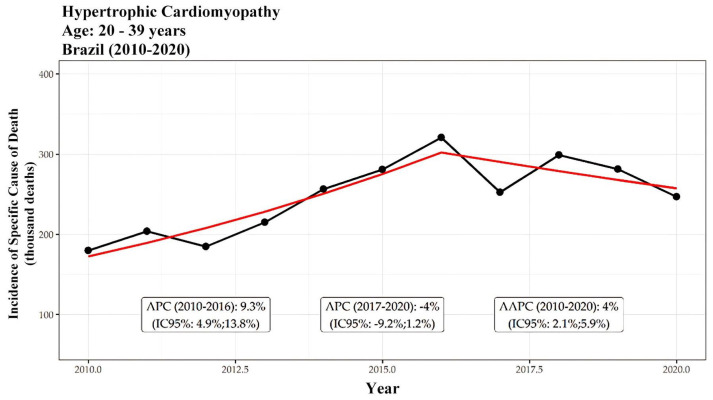
Incidences of specific cause of death per thousand inhabitants related to hypertrophic cardiomyopathy are presented for patients aged 20 to 39 years. The black line represents the incidence, while the red line reflects the segmented model, which generates the APC and AAPC. APC, or annual percent change, indicates the average annual variation. Two statistically significant trends are observed: growth of 16% per year from 2010 to 2016 and a decrease of −9.2% per year from 2017 to 2020. The AAPC was 5.7%.

## Data Availability

The data used are available at https://datasus.saude.gov.br/mortalidade-desde-1996-pela-cid-10 (accessed on 10 March 2024).

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
