# Peer review of "Mortality from Hypertrophic Cardiomyopathy in Brazil—Historical Series"

_ijerph, 2024, doi:10.3390/ijerph21111498_

Round 1
Reviewer 1 Report
Comments and Suggestions for Authors
Title: Mortality from Hypertrophic Cardiomyopathy in Brazil Historical Series.
Reviewer Summary: Left ventricular hypertrophy is the hallmark of HCM and affects both sexes. However, the estimate of mortality rate in our region is hampered by underdiagnosis, socioracial variables, and shortcomings in healthcare. Thus, the main goal of this study was to assess the mortality trend from 2010 to 2020 in Brazil related to HCM. The SIM, created by the Ministry of Health, provides secondary data for this quantitative ecological time-series analysis. The AAPC and the APC were used for the analysis of death incidence and trend evaluation. According to this study, Caucasians 40 years of age or older accounted for most of all deaths caused by HCM. Between 2010 and 2018, there was also a noticeable upward surge in the number of HCM-related deaths among Caucasians and brown individuals who were 40 years of age or older. The northeastern and southeast regions of the country experienced a higher increase in the death rate during this time, as did women. Health system managers must thus make the efforts to find out the factors that impact HCM-related death, as well as to support healthcare facilities in developing clinical protocols that facilitate early diagnosis and establish suitable treatment with the goal of enhancing the prognosis and quality of life of patients.
Weakness:
1. There is a spelling mistake in the Title. Cardiomyopathy not cardiomiopathy.
2. This study's primary drawback was its inability to differentiate reports of HCM-related mortality from those of other heart disorders that shared comparable phenocopies, or similar heart morphologies.
3. Figures quality needs to improve.
4. Why there was an increase in incidence of HCM in women between 2010-2018?
5. The drawbacks of these types of ecological time series studies are that there will be a variation of information and quality due to source diversity.
6. Another drawback of these time series studies includes human error. This may lead to the incorrect data model being identified, which could have a domino effect on the result. It could also be challenging to get the relevant data points.
Reviewer 2 Report
Comments and Suggestions for Authors
Mortality from Hypertrophic Cardiomiopathy in Brazil – Historical Series by Emerson de Santana Santos.
Comments:
Abstract:
- The reviewer suggests rephrasing the study objective to be clearer. It is implied that the study is meant to “evaluate the trend of mortality due to HCM,” but this could be stated more explicitly and concisely as: “This study aimed to assess the mortality trends associated with HCM in Brazil from 2010 to 2020, with a focus on socioracial factors and healthcare disparities.”
- The reviewer suggests rephrasing the phrase “white and brown men and women aged 40 years or older”. Instead of “brown,” can you consider using a more specific racial classification, depending on how Brazil categorizes race in health studies.
- The results section of the abstract is a bit vague and lacks specific quantitative data. It would be helpful to include percentages reflecting the increase in mortality rates and to specify the period (2010–2018). For example: “The incidence of deaths due to HCM increased by X% in females and by Y% in the northeastern and southeastern regions from 2010 to 2018.”
- The reviewer suggests improving the conclusion of the abstract. Rather than saying “it is mandatory for health system managers to invest in recognizing the variables,” rephrase to make it more direct: “Health system managers must prioritize the identification of factors influencing HCM mortality and implement clinical protocols to improve early diagnosis and treatment, thereby enhancing patient survival and quality of life.”
Introduction:
- The reviewer suggests providing more context about why sociodemographic factors are being evaluated. For example, mention how socioeconomic status, access to healthcare, and racial or ethnic disparities could influence mortality trends, particularly in a diverse country like Brazil. This addition would create a stronger rationale for the study.
- The transition from discussing the underdiagnosis of HCM to the management advances could be smoother. Consider adding a linking phrase like: “Despite these challenges, significant progress in the management of HCM has been made in recent decades...” This would help bridge the gap between underdiagnosis and advances in care.
- The introduction would benefit from a brief mention of why studying HCM in Brazil is important. Include any relevant information on how healthcare disparities, socioracial diversity, or regional differences in Brazil might influence HCM diagnosis and outcomes.
- The reviewer suggests minor language adjustments to improve clarity and conciseness. For example, in line 24, it is recommended to change “with development of cardiac muscle fibrosis” to “leading to cardiac muscle fibrosis.” In line 26, the phrase “and arrhythmias, such as atrial fibrillation” could be rephrased as “and arrhythmias, including atrial fibrillation.” Additionally, to avoid redundancy in line 48, the reviewer suggests revising the sentence to: “These advances, including rhythm monitoring, pharmacotherapy, and risk stratification, have contributed to the decline in mortality.”
Materials and Methods:
- The reviewer suggests rephrasing the description of the sample to be more concise. For example: “The sample included all deaths recorded in the SIM platform from 2010 to 2020 due to general cardiomyopathy or HCM, categorized by sex, age group (0-19, 20-39, 40-59, 60-79, and 80+ years), region, and race.”
- The statement on ethical compliance can be rephrased for clarity. Consider:
“As the dataset is publicly accessible, contains no personal identifiers, and does not compromise participant confidentiality, the study was exempt from review under National Health Council Resolutions 466/2012 and 510/2016.”
- The reviewer suggests being more precise about how the data were processed and analyzed. Example: “Data were tabulated using TABWIN (DATASUS) software, then exported to Microsoft Excel 2019 for further statistical analysis.”
- Avoid redundancy in the description of APC/AAPC by integrating the explanation with trend analysis: “Annual Percent Change (APC) and Average Annual Percent Change (AAPC) were calculated to identify trends in mortality over time, with APC estimated for each segment if inflection points were detected.”
- It would be helpful to describe how missing or incomplete data (such as the "ignored" category) were treated in the analysis. Were these records excluded, analyzed separately, or included in a sensitivity analysis?
- Instead of “The significance level adopted was 5%,” consider:
“Statistical significance was set at p < 0.05.”
Results:
- When reporting trends, it would be useful to clarify the statistical significance of the observed trends (e.g., p-values). This will strengthen the interpretation of results.
- The section mentions both increases and decreases in trends but concludes that there is overall “stagnation” across many categories. Consider clarifying this conclusion with more nuanced language (e.g., “Despite fluctuations in specific periods, the overall trend remained relatively stable over the decade.”).
- The reviewer recommends subdividing the Results section into key themes to enhance clarity and readability. For example: Sex Trends: Summarize male and female mortality trends separately, highlighting significant similarities and differences between the two groups. Age Trends: Focus on age groups with the highest mortality rates (40–59 and 60–79) and provide further detail on the stagnation observed in younger age groups. Race Trends: Offer a brief interpretation of the disparities among white, brown, and black populations, emphasizing key differences. Regional Trends: Highlight regions with the most notable changes, such as the consistent decline in the North and the growth observed in the Southeast.
Discussion:
- While the reduction in deaths from HCM between 2019 and 2020 is attributed to the pandemic, more explanation is needed on how COVID-19 might have led to underreporting or affected access to care. Providing a more detailed analysis of how cardiovascular deaths were likely underreported or misclassified during the pandemic would strengthen the conclusion.
- When discussing the increasing trend in female mortality, the study mentions potential factors like delayed diagnosis and heart failure, but there isn’t a definitive conclusion on why women have higher mortality. It would be useful to propose stronger explanations or explore whether biological, social, or healthcare access factors contribute more to this discrepancy.
- The discussion mentions the under-referral of black patients but does not fully explore why black patients showed the highest rate of increase in mortality until 2015. Delving into specific healthcare disparities, such as genetic testing rates, and access to advanced treatments like implantable cardioverter-defibrillators (ICDs) would add depth to this finding.
- The section briefly touches on mortality differences across age groups but doesn’t fully explore why individuals aged 20–39 showed steady growth while younger age groups stagnated. Expanding on the possible reasons for stagnation in younger populations, such as advancements in early detection, might provide more insight.
- The discussion on regional differences, especially the higher concentration of deaths in the Southeast and Northeast, could benefit from more context. It would be helpful to include more detailed information about healthcare infrastructure in these regions and how this may influence mortality outcomes.
- The discussion presents trends (e.g., the stagnation in the 0–19 age group or the sharp drop in mortality between 2019 and 2020) but doesn’t always provide a thorough interpretation. Brief explanations of why these trends might have occurred (e.g., improvements in pediatric care or disruptions during the pandemic) would enhance the discussion.
Conclusion:
- It would be beneficial to mention how your study findings could guide public health efforts. For example, suggest the need for targeted interventions in high-risk populations (e.g., women, specific racial groups, and certain regions).
- While the pandemic’s impact is mentioned, the conclusion could provide more clarity. For example, suggest specific measures to improve data accuracy during future public health crises, such as strengthening reporting systems.
Comments on the Quality of English LanguageMinor editing of English language required.
Round 2
Reviewer 2 Report
Comments and Suggestions for Authors
Thank you for addressing the comments and making the necessary adjustments.
Comments on the Quality of English LanguageMinor editing of English language required.